# Itch in Hidradenitis Suppurativa/Acne Inversa: A Systematic Review

**DOI:** 10.3390/jcm11133813

**Published:** 2022-06-30

**Authors:** Puneet Agarwal, Snehal Balvant Lunge, Nandini Sundar Shetty, Priyanka Karagaiah, Steven Daveluy, Alex G. Ortega-Loayza, Thrasyvoulos Tzellos, Jacek C. Szepietowski, Christos C. Zouboulis, Stephan Grabbe, Mohamad Goldust

**Affiliations:** 1Department of Dermatology, SMS Medical College and Hospital, Jaipur 302004, Rajasthan, India; doc.puneetagarwal@gmail.com; 2Department of Dermatology, Venereology and Leprosy, Bharati Vidyapeeth (DTU) Medical College and Hospital, Pune 411043, Maharashtra, India; drsnehallunge@gmail.com (S.B.L.); ndishetty13@gmail.com (N.S.S.); 3Department of Dermatology, Bangalore Medical College and Research Institute, Bangalore 560002, Karnataka, India; pri20111992@gmail.com; 4Department of Dermatology, Wayne State University, Detroit, MI 48202, USA; sdaveluy@med.wayne.edu; 5Department of Dermatology, Oregon Health and Science University, Portland, OR 97239, USA; ortegalo@ohsu.edu; 6Department of Dermatology, NLSH University Hospital, 8092 Bodø, Norway; ltzellos@googlemail.com; 7European Hidradenitis Suppurativa Foundation e.V., 06847 Dessau, Germany; christos.zouboulis@mhb-fontane.de; 8Department of Dermatology, Venereology and Allergology, Wroclaw Medical University, 50-367 Wroclaw, Poland; 9Departments of Dermatology, Venereology, Allergology and Immunology, Dessau Medical Center, Brandenburg Medical School Theodor Fontane and Faculty of Health Sciences Brandenburg, 06847 Dessau, Germany; 10Department of Dermatology, University Medical Center Mainz, 55131 Mainz, Germany; stephan.grabbe@unimedizin-mainz.de

**Keywords:** hidradenitis suppurativa, acne inversa, pruritus, itch, itching, systematic review

## Abstract

Hidradenitis suppurativa/acne inversa (HS) is a chronic inflammatory disease of the pilosebaceous unit leading to formation of painful, inflammatory nodules, abscesses and tunnels in apocrine gland-bearing areas of the skin. Pain and drainage are the most important symptoms associated with reduction of quality of life in HS. On the other hand, an overlooked symptom in quality of life studies is itch, despite the fact that several studies have reported its importance. Various theories have tried to explain the pathogenesis of itch in HS, such as the presence of mast cells in the cell infiltrates and elevated Ig E levels in the lesional skin. Smoking and advanced stage of disease have been found to be associated with increased intensity of itch. A PUBMED search was conducted to perform a systematic literature review using the term “hidradenitis suppurativa” [all fields], the keywords “pruritus”, “itching”, “itch” [all fields] and with “AND” as operator. Mast cells and mTor signaling were found to be raised in both lesional and perilesional skin. Itch as a presenting symptom has been found in 35–82.6% of patients across multiple studies. It often co-presents with pain and may be misinterpreted as burning, stinging, tickling, tweaking, prickling, etc. The presence of itch is associated with reduced quality of life, depression and impairment of social life. Brodalumab, a monoclonal antibody against IL-17A receptor, produced significant improvements in itch, pain, QoL and depression in patients with moderate to severe HS. Statins have shown some reduction in itch intensity score. Further studies are required to gain a better understanding of the etiopathogenesis and optimal therapeutic modalities for itch in HS that will allow clinicians to better address issue and reduce its impact on quality of life.

## 1. Introduction

Hidradenitis suppurativa/acne inversa (HS) is a chronic debilitating inflammatory skin disease of the pilosebaceous unit [1]. It is characterized by recurrent nodules, abscesses and draining tunnels, which lead to scarring, and occurs in areas rich in apocrine glands [2]. It is a commonly under-diagnosed and under-treated disease [3]. Due to its chronic and relapsing course, it has a grave impact on quality of life (QoL) [4]. Dysregulated immunity, genetic predisposition, smoking and obesity are some of the risk factors found to initiate perifollicular inflammation, which ultimately leads to the formation of painful inflamed lesions [5,6].

The clinical features reported to most reduce the QoL of HS patients are pain and malodorous purulent discharge from skin tunnels. In a study conducted on 1795 subjects, pain was found in 83.6% cases [7]. Few authors, however, have reported in detail prodromal symptoms, such as a burning or stinging sensation, itch, warmth and/or hyperhidrosis [8]. Amongst these prodromal symptoms, itch has been described as an important HS symptom in various studies and has long been overlooked. Thus, we performed a systematic review of the literature to find the prevalence, severity and probable causes of itch in HS.

## 2. Method

A PUBMED search was done for systematic review of literature using the term “hidradenitis suppurativa” [all fields]; the keywords “pruritus”, “itching” and “itch” [all fields]; and “AND” as operator.

The search yielded 105 articles; whereas after the removal of duplicates, 48 articles were left. A total of 28 articles were excluded due to eligibility. Finally, 20 articles were selected for the study. The selection of articles was performed independently by SBL and NSS. All articles were reviewed by the senior author, VRS, who also had the final decision in case of disagreement between the initial selectors (Figure 1 and Table 1).

## 3. Epidemiology

The presence of itch in patients with HS ranged from 62.1% to 77.5% in the selected studies [9,19,25]. The severity of itch has also been assessed in these studies, mostly through a 11-point numerical rating scale (NRS) ranging from 0 to 10, with 0 representing no itch and 10 representing severe itch affecting daily functions and sleep. The NRS score among the patients varied from 4.5 ± 3.5 to 5.4 ± 2.5 points, indicating that the itch in HS is mostly mild to moderate [14,25]. Matusiak et al. and Fernandez et al. found the intensity of itch to be positively correlated with smoking and severity of disease, while, in Molina-Levya et al., the correlation between the two was statistically insignificant [9,14,19]. One study also observed positive association with female sex, number of affected sites and intensity of drainage. In the same study, the presence of Crohn’s disease and acneiform eruptions were also found to increase itch. However, if the patients were on statin treatment, itch seemed to be less intense [14].

Itch in HS has also been found to adversely affect QoL. In a study by Riis et al., a significant association was found between loss of utility and itch [26]. A study by Molina-Leyva et al. on 233 patients also observed that itch caused QoL impairment [14]. Vossen et al. noted that itch affected sleep in 70% and daily activities in 53% of HS patients [25]. A study assessing the influence of itch and pain in HS by using the Athens Insomnia Scale (AIS) and Pittsburgh Sleep Quality Index (PSQI), found that sleep disturbance was more common in patients than insomnia. It affected sleep latency, habitual sleep efficiency, sleep disturbances, sleep duration and daytime dysfunction compared with controls [12,15,21]. In another study with 103 patients with HS, 97% had itching and 62% had pain, and 60% of patients had both symptoms together. A positive correlation was found between itching and impaired QoL. Itch was associated with significant influence on frequency of insomnia [17].

## 4. Etiopathogenesis

The exact pathogenesis of itch in HS is not understood. In the studies by the Rotterdam group (van der Zee et al. [29]; Vossen et al. [25]) histopathological observations were reported in 10 specimens obtained after the surgical excision of HS lesions. They found that tryptase-positive mast cells were significantly increased in both lesional and perilesional skin. They hypothesized the increased number of mast cells to be a possible cause for the reported itch, along with increased serum IgE levels, which may lead to the degranulation of the mast cells, releasing histamine and other mediators, such as proteases.

De Vita et al. [22]. commented that alteration in mammalian target of rapamycin (mTOR) signaling through the mTORC1 pathway could be an alternate hypothesis. Raised levels of mTOR both in lesional and non-lesional skin have been demonstrated in HS, with statistically significant correlation between mTOR gene expression and disease severity [30].

Pascual et al. [31]. measured the level of IgE antibodies in 99 patients with HS. They found elevated levels of IgE in 37.4% patients, with a mean of 186.4 IU/mL and up to a maximum of 2379 IU/mL. The levels were found to be higher in smokers. They concluded that the elevated IgE and an increased infiltration of mast cells in HS could trigger degranulation, thereby releasing histamine and causing itch.

List et al. studied biopsy specimens from HS lesions and perilesional skin from 34 HS patients. They found a larger number of mast cells in lesional skin, with Toluidine blue and CD117 staining, which correlated with itch disease severity by Sartorius score when adjusted to age and sex. They also found a positive correlation between mast cell count and HS activity, suggesting an important relationship between mast cells and HS [16]. Mast cells acts synergistically with other immune cells and secrete multiple proinflammatory cytokines [14,16].

Vossen et al. [25] found an influx of eosinophilic granulocytes, along with a perineural infiltrate comprised of neutrophils and lymphocytes in both lesional and perilesional skin. They observed small fiber neuropathy due to scar formation, suggesting neurogenic inflammation as a probable factor causing itch. This theory of perineural infiltrate and presence of mast cells causing itch was also supported by Matusiak et al. [19].

Mediators of pain and itch overlap in chronic inflammatory conditions [20,32]. Fernandez et al. [9] observed that 74.9% of HS patients experienced both pain and itch. They suggested that, since C-fibres carry sensation for both itch and pain via lateral spinothalamic tract to the thalamus, patients might not be able to accurately distinguish both sensations.

A-fibres, which are sensitive to histaminergic and non-histaminergic stimuli, can be responsible for the induction of itch in HS [22]. A study done by Obara et al. in mice for generalized itch found small diameter A-delta nociceptors with activated mTOR. This activated form of mTOR maintained the excitability of the A-fibers. The inhibition of mTORC1 pathway led to inhibited itch stimuli in mice [33]

Chemokine CCL-26 (eotaxin-3) has recently been identified as an important mediator of inflammation in HS. It has also been found to be significantly elevated in atopic dermatitis and cutaneous T-cell lymphoma, wherein it is likely to mediate the infiltration of eosinophils, basophils and T cells. Both conditions are also associated with high itch scores [18,34].

## 5. Clinical Features

Itch is an important symptom in HS [17]. In several studies, itch was reported to be more intensive in the inguinal region and buttocks than in the axillae. In a study on perianal HS, itch was registered in 10% of patients [35]. The characteristics of itch were studied in detail by Matusiak et al. [19]. They observed that 34.9% patients experienced itch every day, 32.6% a few times a week and 30.2% experienced itch once a month. Itch lasted for less than 10 min in 79.1% of the patients. The primary sensation of itch was described as burning (46.5%), stinging (25.6%), tickling (18.6%), tweaking (16.3%) and prickling (14%) and so on. Excoriation marks were seen in 14% of the patients. Itching was present in the morning and afternoon in 16.3% of the patients, the evening in 51.2% and the night in 18.6%. Sweating, heat, hot bath and physical activity exacerbated itch.

In a retrospective data analysis conducted on 145 patients, itch was reported by 82.1% of the patients and correlated with Hurley stage [28]. Vossen et al. [25] found a similar correlation with Hurley stages along with the number of inflamed areas. In contrast, Pascual et al. [13] found a correlation with active smoking but not with Hurley stages or number of inflamed areas.

In a study evaluating the prodromal symptoms in HS, 20% patients experienced itching at least 24 h before the onset of any visible signs [23]. Through a questionnaire, 26.3% of patients with umbilical flares of HS experienced erythema and itching with no discharge, while 73.7% experienced malodorous or bloody discharge from the umbilicus [24].

Esmann et al. [11] studied the psychosocial impact of HS and found that pain, itching, odor and scars led to irritation, anger, sadness, worry and depression. Odorous draining lesions, severe pain and itch limited patients’ social interaction.

## 6. Treatment

Studies to assess the efficacy of various treatment modalities in HS associated itch are yet to be conducted. Few authors have associated pain with itch, and the alleviation of pain may reduce itch [9]. Many topical, as well as oral, therapies have been suggested to manage pain in HS [27]. Itch can be associated with abscesses and nodules; thus treatment of these lesions may alleviate itch [21].

Brodalumab, a monoclonal antibody against IL-17A receptor, produced significant improvements in itch, pain, QoL and depression in patients with moderate to severe HS [36]. Statins, used for controlling dyslipidemia, have shown some reduction in itch intensity score, probably due to their anti-inflammatory properties and warrants further evaluation [14].

JAK inhibitors have recently been used in treating HS successfully. INCB54707, a experimental molecule which is 52 times more selective for JAK1 than JAK2, has been tried in two multicentric phase 2 studies, wherein a single dose of 90mg oral treatment resulted in a hidradenitis suppurativa clinical response (HiSCR) (defined as at least a 50% reduction in inflammatory lesion count with no increase in abscesses or draining fistulae compared to baseline) score of 88% at week 8. A dose-dependent significant improvement in hidradenitis suppurativa quality of life scores was documented in response to the JAK1 inhibitor [10].

Recent studies have shown that JAK inhibitors can reduce itch symptoms in patients with atopic dermatitis. It is suggested that JAK inhibitors act by blocking neuronal JAK1 signaling in non-inflammatory settings. JAK inhibition with tofacitinib were tried in five patients with severe chronic inflammatory pruritus off-label; all patients showed reduced itch scores, despite having failed to respond to multiple other treatments. These findings suggest the potential use of JAK inhibitors in alleviating itch symptoms in HS [37].

Studies have demonstrated the efficacy of metformin in treating recalcitrant HS. Metformin at a dose of 500 mg 3 times daily for 24 weeks resulted in an average reduction in Sartorius score by 12.78 (*p* = 0.0001) at 24 weeks, which was statistically significant. Improvement in DLQI was significant in 64% of patients. In addition, depression, which was earlier noted to be severe in 11 patients, became nonsevere in 7 of the 11 patients [38]. Arun and Loffeld reported a case of HS with type 2 diabetes that responded exceedingly well to metformin. After four months of treatment, the sinus tracts and leaking abscesses on the patient’s left axilla showed marked resolution, with considerable pain relief [39]. The mechanism by which metformin acts in HS is still not known. However, metformin is known to activate the AMPK pathway through LKB1, eventually causing the inhibition of the mTOR pathway. This pathway is a key factor in mediating itch in pruritus, and metformin may additionally improve the DQLI of patients by itch alleviation [40].

## 7. Conclusions

Despite a lack of complete understanding of itch in HS, it is an important aspect of the disease, reported in 62% to 75% of patients. Further study to gain a better understanding of the etiopathogenesis and optimal therapeutic modalities for itch in HS will allow clinicians to better address this aspect of the disease and reduce its impact on quality of life.

## Figures and Tables

**Figure 1 jcm-11-03813-f001:**
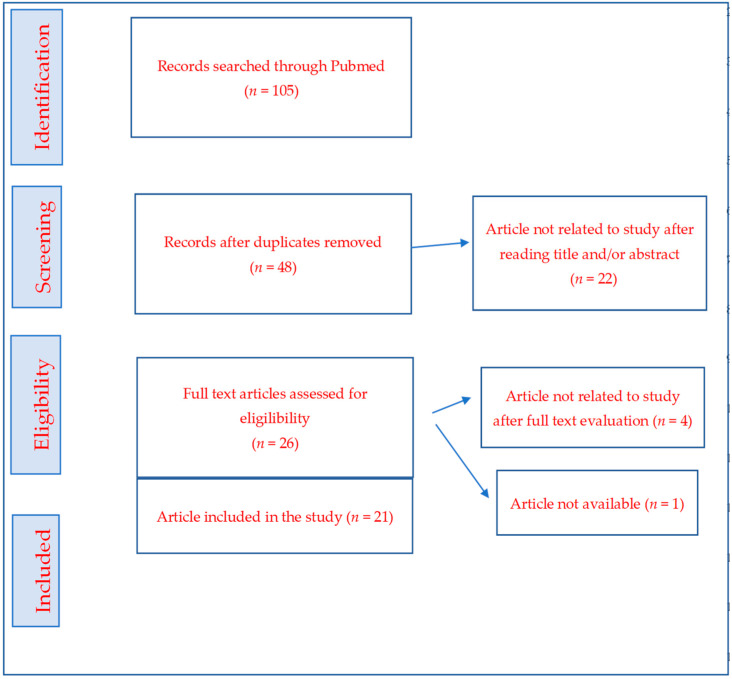
PRISMA flowchart of the selected studies, with criteria and numbers of studies.

**Table 1 jcm-11-03813-t001:** Eligible articles based on inclusion criteria.

Sr.No	PMID	Name of Article	Primary Findings	Author, Ref.	Journal	Year
**1**	**32662883**	Itch and pain by lesion morphology in hidradenitis suppurativa patients	n = 856 Nearly all (95.2%) participants experienced pain, and 77.5% experienced itch. Pain was more bothersome than itch in 70.7%. Discomfort from HS most commonly interfered with exercise (76.6%). Itch was more prevalent in smokers than nonsmokers (85.7% vs. 72.9%, respectively, *p* < 0.0001) and in Hurley stages II and III than stage I (9.2%, 47.8% and 43.1% for Hurley stages I, II and III, respectively, *p* < 0.0001).	Fernandez JM, Rizvi OH, Marr KD, Price KN, Hendricks AJ, Hsiao JL, Shi VY. [9]	Int J Dermatol. 2021 Feb;60(2):e56-e59. doi: 10.1111/ijd.15037. Epub 2020 Jul 14.	2021
**2**	**32416208**	The effect of subcutaneous brodalumab on clinical disease activity in hidradenitis suppurativa: An open-label cohort study	N = 10 No grade 2/3 adverse events associated with the use of brodalumab were reported. A total of 100% patients achieved HiSCR, and 80% achieved IHS4 category change at Week 12. Significant improvements were seen in pain, itch, quality of life and depression.	Frew JW, Navrazhina K, Grand D, Sullivan-Whalen M, Gilleaudeau P, Garcet S, Ungar J, Krueger JG. [10]	J Am Acad Dermatol. 2020 Nov;83(5):1341-1348. doi: 10.1016/j.jaad.2020.05.007. Epub 2020 May 13.	2020
**3**	**32206815**	Umbilical hidradenitis suppurativa symptoms: a case series and review of the literature	N = 19 52.6% (10/19) of the HS patients experienced umbilical symptoms in relation to HS flares. 73.7% (14/19) of the patients experienced malodorous or bloody discharge from the umbilicus. 26.3% (5/19) experienced erythema and itching but no discharge.	Kjærsgaard Andersen R, Jemec GBE, Saunte DM. [11]	Acta Dermatovenerol Alp Pannonica Adriat. 2020 Mar;29(1):3-6.	2020
**4**	**32104123**	An Update on Health-Related Quality of Life and Patient-Reported Outcomes in Hidradenitis Suppurativa	HS patients had higher DQLI scores compared to control population.	Mac Mahon J, Kirthi S, Byrne N, O’Grady C, Tobin AM. [12]	Patient Relat Outcome Meas. 2020 Feb 10;11:21-26. doi: 10.2147/PROM.S174299. eCollection 2020.	2020
**5**	**33054933**	Physical symptoms and psychosocial problems associated with hidradenitis suppurativa: correlation with Hurley stage	N = 145 Hurley stage III patients had significantly higher mean Dermatology Life Quality Index (DLQI) scores (20.2) compared to patients with Hurley stage I (11.3) and II (13.9), (*p* < 0.001 and *p* = 0.001, respectively). >75% of patients reported physical symptoms of drainage, irritation, pain, itching, bleeding and odor. Symptom severity was most strongly correlated with disease severity for odor (correlation coefficient 0.4, *p* < 0.001), difficulty moving arms (0.323, *p* < 0.001), negative impact on job/school (0.303, *p* < 0.001) and negative impact on relationships (0.298, *p* < 0.001)	McKenzie SA, Harview CL, Truong AK, Grogan TR, Shi VY, Bennett RG, Hsiao JL. [13]	Dermatol Online J. 2020 Sep 15;26(9):13030/qt4rm8w7kn.	2020
**6**	**31466061**	Pruritus and Malodour in Patients with Hidradenitis Suppurativa: Impact on Quality of Life and Clinical Features Associated with Symptom Severity	N = 233 Both pruritus and malodor positively correlated with worse quality of life (*p* < 0.05). Pruritus intensity was associated with the number of regions affected by HS, female sex, the intensity of suppuration and the presence of comorbid Crohn’s disease. Statin use was associated with lower levels of pruritus.	Molina-Leyva A, Cuenca-Barrales C. [14]	Dermatology. 2020;236(1):59-65. doi: 10.1159/000502139. Epub 2019 Aug 29.	2020
**7**	**31997991**	Sleep quality among adult patients with chronic dermatoses	N = 108 HS patients 50 controls, Pittsburgh Sleep Quality Index (PSQI) mean scores assessed as 6.5 ±3.6 points and 3.1 ± 1.9 points, respectively (*p*< 0.0001). Pain seems to play a crucial role in impairing sleep quality in HS patients.	Kaaz K, Szepietowski JC, Matusiak Ł. [15]	Postepy Dermatol Alergol. 2019 Dec;36(6):659-666. doi: 10.5114/ada.2019.84007. Epub 2019 Apr 9.	2019
**8**	**30877368**	Mast cells in hidradenitis suppurativa: a clinicopathological study	N = 34 Mast cells (MC) were present to a greater degree in HS-lesions than in perilesional skin (*p* = 0.004). Disease severity (Sartorius score) was correlated to with MC count and itch when adjusted for sex and age (*p* = 0.042). A positive correlation between MC count and HS activity was detected, suggesting a potential link between MC and HS.	List EK, Pascual JC, Zarchi K, Nürnberg BM, Jemec GBE. [16]	Arch Dermatol Res. 2019 May;311(4):331-335. doi: 10.1007/s00403-019-01910-3. Epub 2019 Mar 15.	2019
**9**	**30924968**	Proceeding report of the third symposium on Hidradenitis Suppurativa advances (SHSA) 2018	Recent studies suggested that pruritus may be an important symptom associated with HS, especially in the prodromal stage. Histopathology reports found an increase in mast cells in all stages of HS (including perilesional skin). The intensity of pruritus correlated positively with impairment of quality of life and has a significant influence on the frequency of insomnia.	Posso-De Los Rios CJ, Sarfo A, Ghias M, Alhusayen R, Hamzavi I, Lowes MA, Alavi A. [17]	Exp Dermatol. 2019 Jul;28(7):769-775. doi: 10.1111/exd.13928. Epub 2019 Apr 29.	2019
**10**	**30421795**	Novel cytokine and chemokine markers of hidradenitis suppurativa reflect chronic inflammation and itch	CCL-26 is a newly identified inflammatory marker that is upregulated in the circulation of HS patients. IL-16, CCL-4, CXCL-10 and CCL-26 as novel and potentially important players in the pathogenesis of HS. The local and systemic upregulation of CCL-26 in HS patients can be linked to the high pruritus score in HS.	Vossen ARJV, van der Zee HH, Tsoi LC, Xing X, Devalaraja M, Gudjonsson JE, Prens EP. [18]	Allergy. 2019 Mar;74(3):631-634. doi: 10.1111/all.13665. Epub 2018 Dec 10.	2019
**11**	**28971209**	Clinical Characteristics of Pruritus and Pain in Patients with Hidradenitis Suppurativa	N = 103 A total of 62.1% (64/103) experienced pruritus. Pruritus severity was assessed as 5.0 ± 2.1 points, 5.5 ± 2.3 points and 4.6 ±1.9 points (for VASmax, NRSmax and the 4-item Itch Questionnaire, respectively). (51.2%) of patients with HS reported having moderate pruritus, and more (65.0%) reported mild pain. Pruritus was observed predominantly in the buttocks area (90% of pruritic lesions) and armpits (83% and 87% of pruritic lesions). Pruritus intensity correlated significantly with DLQI (r = 0.45, *p* = 0.004; r = 0.48, *p* = 0.002 for VAS and NRS, respectively).	Matusiak Ł, Szczęch J, Kaaz K, Lelonek E, Szepietowski JC. [19]	Acta Derm Venereol. 2018 Feb 7;98(2):191-194. doi: 10.2340/00015555-2815.	2018
**12**	**29363099**	Hidradenitis suppurativa, a review of pathogenesis, associations and management. Part 2	-	Vekic DA, Cains GD. [20]	Australas J Dermatol. 2018 Nov;59(4):261-266. doi: 10.1111/ajd.12766. Epub 2018 Jan 23.	2018
**13**	**29756157**	Influence of Itch and Pain on Sleep Quality in Patients with Hidradenitis Suppurativa	N = 103 Controls = 50 A total of 61% (66/108) experienced itch symptoms. Itch intensity was assessed as 4.1 ± 2.9 and 5.0 ± 2.1 points (for VASmean and VASmax, respectively). No statistically significant differences were found in Athens Insomnia Scale (AIS) scores for patients with HS and controls. The mean scores for Pittsburgh Sleep Quality Index (PSQI) were 6.5 ± 3.6 points (range 0–18) and 3.1 ± 1.9 points (range 0–7) for patients with HS and control subjects, respectively (*p* < 0.0001). HS do not have significantly more frequent insomnia (AIS) but do have significantly more sleep disturbances (PSQI) than controls.	Kaaz K, Szepietowski JC, Matusiak Ł. [21]	Acta Derm Venereol. 2018 Aug 29;98(8):757-761. doi: 10.2340/00015555-2967.	2018
**14**	**28755065**	Comment on: “Assessing Pruritus in Hidradenitis Suppurativa: A Cross-Sectional Study”	mTOR has been found to be increased in the lesional as well as non-lesional skin of HS patients. Moreover, mTOR gene expression statistically correlates with the severity of HS. Such altered mTOR signalling might contribute to explain itch in HS patients, since it has been shown that the mTORC1 pathway plays an important role in itch signalling. If proven oral metformin and other drugs inhibiting mTOR could be used to effectively treat HS associated itch.	De Vita V, Matusiak Ł, Szepietowski JC. [22]	Am J Clin Dermatol. 2017 Oct;18(5):707-708. doi: 10.1007/s40257-017-0314-9.	2017
**15**	**28597178**	Comment on: “Assessing Pruritus in Hidradenitis Suppurativa: A Cross-Sectional Study”	n = 191 The prevalence (NRS score C3) of pruritus was 58.6% (112/191), with a mean pruritus NRS score of 6.2 (±2.3) in this subgroup. The mean intensity of itch in all 191 patients was rated at an NRS score of 3.9 (±3.3). There was significant association with active smoking (OR 1.1–3.8; *p* = 0.02) but not with Hurley stage or number of inflamed areas, both of which had an association with pruritus in the cohort of Vossen et al.	Pascual JC, Alvarez P, Encabo B, González I, Hispán P, Poveda I, Romero D. [23]	Am J Clin Dermatol. 2017 Oct;18(5):705-706. doi: 10.1007/s40257-017-0304-y.	2017
**16**	**28194809**	Prodromal symptoms in hidradenitis suppurativa	N = 72 A total of (83.3%; n = 60) confirmed that they experienced one or more symptom(s) prior to the development of inflamed nodules or abscesses. These included: fatigue (32%), malaise (defined as a fever-like sensation) (23%), headache (11%) and nausea (2%). Localized symptoms included skin erythema (75%), paraesthesia (63%) and itching (20%). Prodromes usually occurred > 24 h (45%) or 12-24 h (20%) before the eruption.	Ring HC, Theut Riis P, Zarchi K, Miller IM, Saunte DM, Jemec GB. [24]	Clin Exp Dermatol. 2017 Apr;42(3):261-265. doi: 10.1111/ced.13025. Epub 2017 Feb 14.	2017
**17**	**28429245**	Assessing Pruritus in Hidradenitis Suppurativa: A Cross-Sectional Study	N = 211 Mean NRS score of 6.1 ± 2.0. Patients with a pruritus NRS score ≥ 3 had more HS-affected body sites than patients with a score < 3 (*p* < 0.001). NRS score ≥ 3 was associated with Hurley III disease (odds ratio [OR] 7.73; *p* = 0.003) and pain (OR 1.34; *p* < 0.001). Pruritus affected sleep and activities of daily living (ADL) in the majority of cases, with an associated modified 5-D itch score of 13.7 ± 3.6 (on a scale from 5 to 25) in 52 HS patients.	Vossen ARJV, Schoenmakers A, van Straalen KR, Prens EP, van der Zee HH. [25]	Am J Clin Dermatol. 2017 Oct;18(5):687-695. doi: 10.1007/s40257-017-0280-2.	2017
**18**	**25940640**	Disutility in Patients with Hidradenitis Suppurativa: A Cross-sectional Study Using EuroQoL-5D	N = 421 A significantly decreased utility in patients with hidradenitis suppurativa was found for all age group levels, except for 65-74-year-olds. The total index score in the cohort was 0.705 (population mean 0.887), and the VAS was 62.25 (population mean 82.6). Multivariate analysis found significant associations between loss of utility and pain, malodor and pruritus (*p* < 0.0001). Patients with hidradenitis suppurativa had a significantly decreased EuroQoL-5D (EQ-5D) compared with the background population.	Riis PT, Vinding GR, Ring HC, Jemec GB. [26]	Acta Derm Venereol. 2016 Feb;96(2):222-6. doi: 10.2340/00015555-2129.	2016
**19**	**21394419**	Psychosocial impact of hidradenitis suppurativa: a qualitative study	N = 12 HS has a great emotional impact on patients and promotes isolation due to fear of stigmatization. Shame and irritation are frequent and relate to smell, scars, itching and pain. Quality of life is adversely affected, and professional support is needed.	Esmann S, Jemec GB. [27]	Acta Derm Venereol. 2011 May;91(3):328-32. doi: 10.2340/00015555-1082.	2011
**20**	**2390907**	Perianal hidradenitis suppurativa. The Lahey Clinic experience	N = 43 Symptoms, including pain, swelling, purulent discharge and pruritus, had been present for a median of six years. Associated medical conditions included diabetes (12%) and obesity (12%), and 70 percent of patients were smokers. Once the correct diagnosis was established, 72% of patients had wide local excision with healing by secondary intention, and 28% of patients had incision and drainage or limited local excision. Although 67% of the patients had recurrence of disease after initial treatment, wide excision was more successful in preventing recurrence.	Wiltz O, Schoetz DJ Jr, Murray JJ, Roberts PL, Coller JA, Veidenheimer MC. [28]	Dis Colon Rectum. 1990 Sep;33(9):731-4. doi: 10.1007/BF02052316.	1990

## Data Availability

Not applicable.

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
