# Peer review of "Itch in Hidradenitis Suppurativa/Acne Inversa: A Systematic Review"

_jcm, 2022, doi:10.3390/jcm11133813_

Round 1

Reviewer 1 Report

Well written and interesting paper that focuses on one of the under-diagnosed and under-estimated symptoms associated to HS. Authors summarize what is known on itch pathogenesis and its clinical features in HS and possible treatment options. Further studies, with special regard to etiopathogenesis, are needed in order to tailor therapeutic approaches to HS.

For this paper, it would be useful a cartoon on possible pathogenic pathways in itch development in HS. 

Author Response

Reviewer 1:

Well written and interesting paper that focuses on one of the under-diagnosed and under-estimated symptoms associated to HS. Authors summarize what is known on itch pathogenesis and its clinical features in HS and possible treatment options. Further studies, with special regard to etiopathogenesis, are needed in order to tailor therapeutic approaches to HS.

For this paper, it would be useful a cartoon on possible pathogenic pathways in itch development in HS. 

R: Thank you for the kind words.  

Reviewer 2 Report

This is a very well-written review on an important topic, HS and Itch. The review covers a good range of publications, tables, and figures. It could be interesting to report the frequency of ACD in HS patients (Patruno C, Fabbrocini G, D'Andrea M, Dastoli S, Marasca C, Napolitano M. Frequency of allergic contact dermatitis in hidradenitis suppurativa patients. Ital J Dermatol Venerol. 2021 Jun;156(3):396-397. doi: 10.23736/S2784-8671.20.06660-2. Epub 2020 Oct 16. PMID: 33070570) and the intriguing link between IL-26 and ACD (Balato A. IL-26 in allergic contact dermatitis: Resource in a state of readiness. Exp Dermatol. 2018 Jun;27(6):681-684. doi: 10.1111/exd.13521. Epub 2018 Apr 10. PMID: 29498775) and the importance of IL-26 in HS pathogenesis (Scala E, Di Caprio R, Cacciapuoti S, Caiazzo G, Fusco A, Tortorella E, Fabbrocini G, Balato A. A new T helper 17 cytokine in hidradenitis suppurativa: antimicrobial and proinflammatory role of interleukin-26. Br J Dermatol. 2019 Nov;181(5):1038-1045. doi: 10.1111/bjd.17854; Scala E, Cacciapuoti S, Garzorz-Stark N, Megna M, Marasca C, Seiringer P, Volz T, Eyerich K, Fabbrocini G. Hidradenitis Suppurativa: Where We Are and Where We Are Going. Cells. 2021 Aug 15;10(8):2094. doi: 10.3390/cells10082094). 

Author Response

Reviewer 2:

This is a very well-written review on an important topic, HS and Itch. The review covers a good range of publications, tables, and figures. It could be interesting to report the frequency of ACD in HS patients (Patruno C, Fabbrocini G, D'Andrea M, Dastoli S, Marasca C, Napolitano M. Frequency of allergic contact dermatitis in hidradenitis suppurativa patients. Ital J Dermatol Venerol. 2021 Jun;156(3):396-397. doi: 10.23736/S2784-8671.20.06660-2. Epub 2020 Oct 16. PMID: 33070570) and the intriguing link between IL-26 and ACD (Balato A. IL-26 in allergic contact dermatitis: Resource in a state of readiness. Exp Dermatol. 2018 Jun;27(6):681-684. doi: 10.1111/exd.13521. Epub 2018 Apr 10. PMID: 29498775) and the importance of IL-26 in HS pathogenesis (Scala E, Di Caprio R, Cacciapuoti S, Caiazzo G, Fusco A, Tortorella E, Fabbrocini G, Balato A. A new T helper 17 cytokine in hidradenitis suppurativa: antimicrobial and proinflammatory role of interleukin-26. Br J Dermatol. 2019 Nov;181(5):1038-1045. doi: 10.1111/bjd.17854; Scala E, Cacciapuoti S, Garzorz-Stark N, Megna M, Marasca C, Seiringer P, Volz T, Eyerich K, Fabbrocini G. Hidradenitis Suppurativa: Where We Are and Where We Are Going. Cells. 2021 Aug 15;10(8):2094. doi: 10.3390/cells10082094). 

R:The articles suggested show an interesting association of HS with ACD. However, the main agenda of our article is to highlight itch in HS. The article suggested (Scala E, Di Caprio R, Cacciapuoti S, Caiazzo G, Fusco A, Tortorella E, Fabbrocini G, Balato A. A new T helper 17 cytokine in hidradenitis suppurativa: antimicrobial and proinflammatory role of interleukin-26. Br J Dermatol. 2019 Nov;181(5):1038-1045. doi: 10.1111/bjd.17854;)  and (Balato A. IL-26 in allergic contact dermatitis: Resource in a state of readiness. Exp Dermatol. 2018 Jun;27(6):681-684. doi: 10.1111/exd.13521. Epub 2018 Apr 10. PMID: 29498775) highlights the role of IL 26 in keratinocyte cytotoxicity and its subsequent effects in ACD. IL26 acts as antimicrobicidal and proinflammatory mediator whose effects may be suboptimal in HS leading to faulty antimicrobicidal activity. However, its role in itch pathways is not clear. IL-26, being a Th17 cell mediator plays an important role in psoriasis, where in it was found to act directly on vascular endothelial cells, promoting proliferation and tube formation, possibly through protein kinase B, extracellular signal–regulated kinase, and NF-κB pathways. Moreover, similar effects of IL-26 were observed in the murine contact hypersensitivity model, indicating that these effects are not restricted to psoriasis. (Itoh T, Hatano R, Komiya E, Otsuka H, Narita Y, Aune TM, Dang NH, Matsuoka S, Naito H, Tominaga M, Takamori K. Biological effects of IL-26 on T cell–mediated skin inflammation, including psoriasis. Journal of Investigative Dermatology. 2019 Apr 1;139(4):878-89.) However, its direct role in causation of itch is unclear at this point. Hence, assuming possible association of IL26 in itch causation at this point would be premature. .

Reviewer 3 Report

This is an interesting and well presented review on itch in hidradenitis suppurativa. I would suggest that the results and conclusion of the review are also briefly presented in the abstract, highlighting the great importance of itch among the clinical symptoms of hidradentitis suppurativa.

Author Response

Reviewer 3:

This is an interesting and well-presented review on itch in hidradenitis suppurativa. I would suggest that the results and conclusion of the review are also briefly presented in the abstract, highlighting the great importance of itch among the clinical symptoms of hidradentitis suppurativa.

R: Thank you for the kind review. Your suggestions have been added.

Reviewer 4 Report

Thank you for an interesting article. The authors provide a relevant introduction to this understudied subject and the findings are structured in a logical manner. However, I do have some comments and questions for you that may improve this manuscript.

L 66: Prevalence when? During flares, prior to flares, or in general?

L 70: I don’t quite understand the NRS interval, as the upper limit 4.5 + 3.5 = 8.0 and 5.4 + 2.5 = 7.9 overlap?!

L 74: Associations or an association with female sex…?

L 79: A significant association

L 132 – 140: This is a relevant description, although I think it could benefit from some guiding comments, i.e. “the frequency of itch was equally distributed as occurring everyday (..%), few times as week (..%), and ….”, “the primary sensations of itch were described as burning (..%) and stinging (..%), but terms as tickling (..%), tweaking (..%), and prickling (..%) were also used.” And so on.

L 134: Prevalence of itch in perianal HS is only 10%, while the overall prevalence is 62.1–77.5%. What are the author’s explanations on this substantial variation?

L 145 – 146: What is the author’s opinion on why itch and inflamed areas or Hurley stage were not correlated?

L 167: Studies?

Table 1: The table would benefit from a column describing the primary findings, and a column indicating the country of origin

Author Response

Reviewer 4:

Thank you for an interesting article. The authors provide a relevant introduction to this understudied subject and the findings are structured in a logical manner. However, I do have some comments and questions for you that may improve this manuscript.

Thank you for your valuable suggestions. Please find the response to all your queries below.

L 66: Prevalence when? During flares, prior to flares, or in general?

R: This refers to prevalence in general, as not many studies have been done on itch in HS.

L 70: I don’t quite understand the NRS interval, as the upper limit 4.5 + 3.5 = 8.0 and 5.4 + 2.5 = 7.9 overlap?!

R: These NRS scores are across multiple studies, and not a single study. It indicates that itch severity is more or less the same in the studies reviewed.

L 74: Associations or an association with female sex…?

R: In a plural context it is ‘association’ and if you are asking with reference to female association, in this context it is association with the female gender.

L 79: A significant association

R: corrected

L 132 – 140: This is a relevant description, although I think it could benefit from some guiding comments, i.e. “the frequency of itch was equally distributed as occurring everyday (..%), few times as week (..%), and ….”, “the primary sensations of itch were described as burning (..%) and stinging (..%), but terms as tickling (..%), tweaking (..%), and prickling (..%) were also used.” And so on.

 R: Noted

L 134: Prevalence of itch in perianal HS is only 10%, while the overall prevalence is 62.1–77.5%. What are the author’s explanations on this substantial variation?

R: The difference may be due to bias in recollecting the symptom as it was a retrospective analysis and itch may not have been the most debilitating symptom and 10% may not be  a true reflection of the actual severity.

L 145 – 146: What is the author’s opinion on why itch and inflamed areas or Hurley stage were not correlated?

R: The difference in findings may be associated with the fact that in study by Vossen et a a clear creiteria of NRS >3 was defined to assess correlation with Hurley stage and number of lesions and whereas in Pascaul et al this may have been missing. Although, we are still unclear of the reason.

L 167: Studies?

R: Grammar seems to be correct in the sentence under review

Table 1: The table would benefit from a column describing the primary findings, and a column indicating the country of origin

R: Primary findings have been added as per you suggestion
